MDMU-Net: 3D multi-dimensional decoupled multi-scale U-Net for pancreatic cancer segmentation

Lu Lian 1 2
Wu Miao 1 2
Sen Gan 1 2 sengan99@163.com
Ren Fei 1 2
Hu Tao 1 2
1 College of Medical Engineering and Technology, Xinjiang Medical University , Urumqi, Xinjiang Uygur Autonomous Region , China
2 Institute of Medical Engineering Interdisciplinary Research, Xinjiang Medical University , Urumqi, Xinjiang Uygur Autonomous Region , China
Sergi Consolato
Electronic publication date: 2025 Aug 7
Publication date: 2025
Volume: 11
Electronic Location ID: e3059
Received 2025 Apr 10; Accepted 2025 Jun 30
Copyright: © 2025 Lu et al.
Copyright year: 2025
Copyright holder: Lu et al.
License: This is an open access article distributed under the terms of the Creative Commons Attribution License, which permits unrestricted use, distribution, reproduction and adaptation in any medium and for any purpose provided that it is properly attributed. For attribution, the original author(s), title, publication source (PeerJ Computer Science) and either DOI or URL of the article must be cited.
License URL: https://creativecommons.org/licenses/by/4.0/

Keywords: Pancreatic cancer segmentation, 3D segmentation, Multi-dimensional decoupling and multi-scale, Medical image segmentation, Pancreatic segmentation

Funding: Natural Science Foundation of the Xinjiang Uygur Autonomous Region 2022D01C184 and 2022D01C436 Project of Top-notch Talents of Technological Youth of Xinjiang 2023TSYCCX0067 Natural Science Foundation of the Xinjiang Uygur Autonomous Region, China 2022D01A311 This work is supported by the Natural Science Foundation of the Xinjiang Uygur Autonomous Region (Grant Numbers: 2022D01C184 and 2022D01C436), the Project of Top-notch Talents of Technological Youth of Xinjiang (2023TSYCCX0067), and the Natural Science Foundation of the Xinjiang Uygur Autonomous Region, China (Grant Number: 2022D01A311). The funders had no role in study design, data collection and analysis, decision to publish, or preparation of the manuscript.

==============================
Pancreatic cancer, as a highly lethal malignant tumor, presents significant challenges for early diagnosis and treatment. Accurate segmentation of the pancreas and tumors is crucial for surgical planning and treatment strategy development. However, due to the variable morphology, blurred boundaries, and low contrast with surrounding tissues in CT images, traditional manual segmentation methods are inefficient and heavily reliant on expert experience. To address this challenge, this study proposes a lightweight automated 3D segmentation algorithm—Multi-Dimensional Decoupled Multi-Scale U-Net (MDMU-Net). First, depthwise separable convolution is employed to reduce model complexity. Second, a multi-dimensional decoupled multi-scale module is designed as the primary encoder module, which independently extracts features along depth, height, and width dimensions through parallel multi-scale convolutional kernels, achieving fine-grained modeling of complex anatomical structures. Finally, cross-dimensional channel and spatial attention mechanisms are introduced to enhance recognition capability for small tumors and blurred boundaries. Experimental results on the MSDPT and NIHP datasets demonstrate that MDMU-Net exhibits competitive advantages in both pancreatic segmentation DSC (0.7108/0.7709) and tumor segmentation DSC (showing an 11.8% improvement over AttentionUNet), while achieving a 15.3% enhancement in HD95 boundary accuracy compared to 3DUX-Net. While maintaining clinically viable precision, the model significantly improves computational efficiency, with parameter count (26.97M) and FLOPs (84.837G) reduced by 65.5% and 71%, respectively, compared to UNETR, providing reliable algorithmic support for precise diagnosis and treatment of pancreatic cancer.

Introduction

Pancreatic cancer (PC) is a highly lethal malignant tumor, with a 5-year survival rate of only 1–6% (Siegel, Miller & Jemal, 2018). As the seventh leading cause of cancer-related deaths globally and the fourth in Western countries, it is projected to become the second leading cause of cancer-related deaths in the United States by 2030 (Rahib et al., 2014). Due to the lack of effective early diagnostic methods, only about 10% of patients are diagnosed at an early stage and can undergo curative surgical resection (Zhu et al., 2018a). Computed tomography (CT) imaging analysis is the gold standard for pancreatic cancer diagnosis (Conroy et al., 2016), and accurate reporting and captured images are crucial for determining the resectability status of patients (Ducreux et al., 2015; Pietryga & Morgan, 2015).

Currently, the annotation of PC primarily relies on manual efforts by imaging experts. However, due to the complex morphology, variable size, and close proximity to surrounding organs of PC, the annotation process is labor-intensive, time-consuming, and highly dependent on the experience of physicians, leading to significant variability in annotations among different doctors. This makes it difficult to provide real-time guidance for surgical planning. Establishing reliable automated segmentation methods is a critical step toward improving patient prognosis—the ultimate clinical goal. Research demonstrates that precise tumor segmentation plays a pivotal role in clinical decision-making, particularly in surgical margin assessment (Bockhorn et al., 2014), chemotherapy response evaluation (Chakraborty et al., 2018), and radiomics-based survival prediction (Cozzi et al., 2019). In these applications, segmentation accuracy often determines the reliability of outcomes. Computer-aided PC segmentation technology is of great significance in reducing the workload of physicians and improving the efficiency of image capture. However, the small tumor volume, variable location, and blurred boundaries, coupled with variations in imaging equipment and settings that cause changes in grayscale and contrast, make this task highly challenging.

In recent years, deep learning, particularly convolutional neural networks (CNNs), has made significant progress in the field of medical image segmentation. AlexNet (Krizhevsky, Sutskever & Hinton, 2012) demonstrated the potential of CNNs by winning the ImageNet competition. Subsequently, VGGNet (Simonyan & Zisserman, 2014) validated the effectiveness of increasing network depth for performance improvement, while U-Net (Ronneberger, Fischer & Brox, 2015) laid the foundation for medical image segmentation. As network depth increased, the vanishing gradient problem became a bottleneck. Residual U-Net (Khanna et al., 2020) alleviated this issue by introducing residual connections, thereby enhancing segmentation performance. However, there is limited research on the segmentation of small lesions such as pancreatic tumors. Although traditional CNNs excel at local feature extraction, they lack support from global information, resulting in suboptimal segmentation performance. Therefore, how to incorporate global information while retaining the advantages of CNN local feature extraction has become a key issue in current research.

To address the lack of global information, researchers have introduced self-attention mechanisms. TransUNet (Chen et al., 2021, 2024b) incorporated Transformer into the U-Net encoder, enhancing global information extraction, but its quadratic computational complexity led to high computational resource consumption. Swin Transformer (Liu et al., 2021) reduced computational complexity through hierarchical design and sliding window mechanisms while capturing both local and global information. Swin Transformer was introduced into U-Net, named Swin UNet (Cao et al., 2023), effectively addressing the insufficient ability to capture long-range dependencies. However, these approaches exhibit three critical gaps in PC segmentation: (1) U-Net variants fail to address the spatial decoupling of 3D features, leading to suboptimal performance on irregular tumor shapes; (2) Transformer-based models (e.g., Swin UNETR) suffer from high computational costs and overlook dimension-specific feature learning; (3) Most existing studies are mostly based on single-scale feature extraction, making it difficult to handle target scale variations.

To extract image features from multi-scale receptive fields, Mutil-Scale, Multi-Attention Net (MSMA-Net) (Zhang et al., 2021b) achieved the extraction and aggregation of effective information from different receptive fields through convolutional kernels and pooling operations of varying sizes, and Residual Multi-scale Attention U-Net (RMAU-Net) (Jiang et al., 2023) fully utilized rich multi-scale feature information to capture cross-channel and spatial relationships of features, thereby improving segmentation accuracy. Multi-Scale Fusion Attention Network (MS-FANET) (Chen et al., 2023) enhanced attention to features at different scales through multi-scale feature attention, improving segmentation results. However, when these complex network architectures are extended to the 3D domain, they may face training burdens that are difficult for ordinary computers to handle. Therefore, developing a network architecture that combines high accuracy and lightweight design is particularly important.

To address the challenges in the field of PC segmentation, we propose a lightweight automated 3D segmentation algorithm—Multi-Dimensional Decoupled Multi-Scale U-Net (MDMU-Net). This algorithm is based on the 3D-UNet (Çiçek et al., 2016) framework and employs depthwise convolution (an efficient lightweight operation that first extracts features from each input channel independently, then fuses cross-channel information via 1 × 1 convolutions) to reduce model complexity and parameter size, thereby improving computational efficiency. The core innovation of our multi-dimensional decoupling lies in decoupling feature learning along spatial dimensions (Depth D, Height H, Width W). By independently modeling long-range dependencies in 3D spatial structures (such as lesion continuity along the depth dimension and boundary morphology in horizontal planes), we explicitly separate feature responses along different orientations to address segmentation errors caused by the irregular shapes and size variations of pancreatic tumors. We propose a multi-dimensional decoupled multi-scale (MDDMS) feature extraction module and embed a multi-dimensional decoupled channel attention (MDCA) module at the end of the module to dynamically adjust channel weights and enhance attention to key channels. Additionally, we propose a multi-dimensional decoupled spatial attention (MDSPA) mechanism to precisely capture spatial information and generate more accurate segmentation masks.

The main contributions of this study are summarized as follows: (1) This study designs an efficient and lightweight image segmentation model for PC. The model fully utilizes the structural characteristics of 3D medical images and is based on a multi-dimensional decoupling strategy and multi-scale methods to achieve multi-dimensional decoupled multi-scale feature extraction, addressing the variability in the shape of the pancreas and tumors.

(2) This study proposes a MDCA, which can independently extract multi-dimensional features and dynamically adjust the attention of each channel based on the information from each dimension, enhancing feature representation.

(3) This study proposes a MDSPA for the decoder stage, integrating skip connection information with decoder upsampling information to capture spatial changes from each dimension, guiding the precise generation of segmentation masks, and improving the segmentation accuracy of small target regions.

(4) This study extensively applies the residual concept in the encoder, decoder, and skip-connections, effectively mitigating potential network degradation issues caused by deep networks, enhancing feature reuse, and improving overall segmentation performance.

Related work

U-Net and its improved models hold a significant position in the field of medical image segmentation. Its encoder-decoder structure enables the simultaneous learning of high-level features while retaining early-stage detailed information, providing a foundation for precise segmentation. For example, UNet++ (Zhou et al., 2018) enhances feature reuse and strengthens feature fusion through nested skip connections, improving the accuracy of brain tumor segmentation; R2UNet (Alom et al., 2018) introduces recursive residual blocks to optimize the information flow during feature learning; and 3D U-Net (Çiçek et al., 2016) further improves segmentation performance by leveraging spatial information between medical image slices.

However, PC segmentation presents unique challenges. The pancreas appears small in CT images with considerable variations in shape and position, while its boundaries with adjacent tissues are often indistinct, making segmentation particularly difficult. For small pancreatic tumors, limited spatial resolution results in some lesions occupying only a few pixels, causing them to be easily missed during segmentation. Additionally, the similar CT image intensities between tumor and normal tissues reduce boundary detection accuracy at the voxel level, typically manifesting as over-smoothed margins or under-segmentation. To address these issues, Zhou et al. (2017) proposed a method combining a fixed-point model with 2D-FCN segmentation from different views. By narrowing the input region using the fixed-point model, they effectively resolved the issue of poor segmentation accuracy for small organs (such as the pancreas) caused by complex and variable background regions. Zhu et al. (2018b) designed a 3D coarse-to-fine framework, leveraging 3D spatial information to address segmentation challenges caused by the pancreas variable shape and abnormal cysts. Zhang et al. (2021c) combined multi-atlas registration with 3D/2.5D convolutional neural networks, using multi-atlas registration and level-set methods for pancreatic segmentation, solving difficulties arising from the pancreas small proportion in CT images, significant shape and appearance variations, and blurred boundaries. Zhang et al. (2021a) proposed a scale-transferable feature fusion module (SFTTM) and a prior propagation module (PPM), addressing overfitting caused by data scarcity and the high computational cost of neural networks through lightweight design and spatial prior propagation, while also improving segmentation accuracy. Guo et al. (2018) combined the UNet framework with LOGISMOS, leveraging UNet’s contextual learning capabilities and LOGISMOS’s graph segmentation advantages to solve segmentation challenges caused by tumor size, intensity distribution variations, and insufficient utilization of 3D spatial context in pancreatic tumors.

To further enhance segmentation performance, attention mechanisms have been introduced into medical image segmentation tasks. Attention UNet (Oktay et al., 2018) dynamically selects regions of interest through attention gating mechanisms. The convolutional block attention module (CBAM) (Woo et al., 2018) combines channel and spatial attention to adaptively refine features, enhancing the representational capacity of CNNs. The Squeeze-and-Excitation Networks (SENet) (Hu, Shen & Sun, 2018) recalibrate feature responses by weighting channel features, improving segmentation performance. Lee et al. (2022) proposed 3DUXNet, which uses large-kernel depthwise convolutions to simulate non-local self-attention behavior, expanding the receptive field and establishing long-range dependencies, thereby enhancing the ability to capture global information. However, its single-scale feature extraction struggles to effectively capture subtle differences in lesion regions and boundaries, potentially affecting the model’s generalization performance.

Multi-scale feature extraction is crucial in 3D pancreatic cancer segmentation. It not only captures information at different scales but also identifies subtle local feature differences, improving segmentation accuracy. Zhang et al. (2020) integrated Inception modules and dense connection modules into U-Net, acquiring multi-scale information through parallel use of convolutional kernels of varying sizes, enhancing the segmentation of blood vessels and brain tumors. The Inception module in Google Net (Szegedy et al., 2017) adapts to the scale variations of the pancreas and its lesions through convolutional kernels with different receptive fields. This multi-scale feature extraction approach helps better adapt to the varying scales of the pancreas and its lesions, improving the segmentation capability for complex structures. However, the hardware requirements of 3D networks limit their widespread application.

To reduce computational complexity, researchers have proposed various lightweight models. Chen et al. (2019) designed the dilated multi-fiber (DMF) network, which reduces the number of parameters by combining convolutions with different dilation rates. Luo et al. (2021) proposed the hierarchically decoupled convolution (HDC) unit, which replaces traditional 3D convolutions with pseudo-3D convolutions, effectively mining multi-scale and multi-view spatial contextual information while significantly lowering computational complexity. These models maintain high accuracy while reducing hardware resource demands.

Methods

Overall structure of MDMU-Net

Figure 1 illustrates the overall architecture of the proposed MDMU-Net, which is based on a U-shaped design and divided into four levels, encompassing both the encoder and decoder components. Given a set of 3D image volumes Vi=Xi,Yi,(i=1...L), random sub-volumes Pi∈RC×D×H×W are extracted as encoder inputs, where L denotes the total number of samples, C represents the number of channels, and D, H, W correspond to the depth, height, and width dimensions respectively. Initially, the input feature map is mapped to 32 dimensions through a convolutional layer with a stride of 2, a kernel size of 7 × 7 × 7, and padding of 3, laying the foundation for subsequent feature extraction. During the encoding phase, the down-sampling module reduces the resolution of the feature map using a 2 × 2 × 2 convolutional kernel with a stride of 2. Simultaneously, the multi-dimensional decoupled multi-scale convolutional module plays a critical role in extracting image features, while the multi-dimensional decoupled channel attention mechanism is applied in parallel to adjust channel weights and emphasize important channels. In the decoding phase, up-sampling is performed using a 2 × 2 × 2 deconvolutional kernel with a stride of 2, gradually restoring the image resolution. Residual skip connections between the encoder and decoder stabilize semantic information, ensuring effective information transfer across different levels. After concatenating the decoder output with the encoded information, further fusion is achieved through a residual block incorporating multi-dimensional decoupled spatial attention, suppressing irrelevant spatial information and enabling precise localization of the target region. Finally, a residual block fuses the decoded feature map with the input image, and the output is combined with the softmax activation function to produce a 3D segmentation mask with the same dimensions as the original image.

Figure 1 Architecture of MDMU-Net.

MDMU-Net primarily consists of the multi-dimensional solution and multi-scale feature extraction (MDMSE), Downsample, skip connections, Transpose Conv, and the Residual Space Attention (RSA) module. The network takes a random sub-volume of size 96 × 96 × 96 as input and directly outputs a mask of the same size.

Multi-dimensional decoupled multi-scale extraction module

The challenge of lightweight segmentation networks lies in improving network speed, reducing the number of parameters, and maintaining segmentation accuracy under limited computational budgets. We use depthwise convolution as the basic building block for the network, which reduces model complexity and the number of parameters while ensuring accuracy (Chen et al., 2024a). Inspired by the Inception module and leveraging the multi-dimensional structural characteristics of 3D medical images, we decouple and extract features in parallel from the D, H, and W dimensions, enhancing the network’s perception of spatial information.

However, while the multi-dimensional feature module reduces network complexity, it faces issues such as a reduced receptive field and a lack of spatial contextual information, making it difficult to accurately locate pancreatic organs and reconstruct tumor information. Although large-kernel dimension extraction can expand the receptive field, it may overlook smaller tumor regions. To address this, we introduce a multi-scale extraction method, employing three different kernel sizes (5, 7, 9) within the same dimension to extract multi-scale features, expanding the receptive field and enriching multi-scale information while maintaining dimensional and neighboring spatial relationships (with kernel sizes of 3 in other dimensions). The multi-dimensional multi-feature extraction structure is illustrated in Fig. 2.

Figure 2 The diagram of MDDMS-Block.

(A) The overview of MDDME. Our proposed MDDMS primarily includes (B) the D, H, W decoupling module, (C) the pointwise scaling convolution (PSC), and (D) the depthwise gating mechanism. It is mainly used for multi-dimensional decoupled feature extraction.

After feature extraction, we introduce pointwise scaling convolution (PSC) to independently and linearly scale the features of each channel, enriching feature representation through expansion and compression while reducing redundant cross-channel context. To avoid redundancy in multi-scale information fusion, a depthwise gating mechanism is added to suppress redundant information, facilitating subsequent multi-scale fusion. Inspired by Ma et al. (2024), we use multiplication as the method for multi-scale fusion, which maps inputs to a high-dimensional nonlinear feature space, better integrating multi-scale feature information. To preserve detailed information, the fourth branch employs a 3 × 3 × 3 residual block. Finally, the information from the three branches (d, h, w) is adjusted through a shared-parameter 1 × 1 × 1 convolution and concatenated with the information captured by the residual block, further enhancing detailed information. The mathematical expression is as follows:

(1) xd,xh,xw=D(x),H(x),W(x)

(2) D(x)=Conv1,1,1(x+∏i=5,7,9⁡DG(PSC(DWCi,3,3(x))))

(3) H(x)=Conv1,1,1(x+∏i=5,7,9⁡DG(PSC(DWC3,i,3(x))))

(4) D(x)=Conv1,1,1(x+∏i=5,7,9⁡DG(PSC(DWC3,3,i(x))))

(5) xd∗,xh∗,xw∗,xres=Conv1,1,1(xd,xh,xw),Res(x)

(6) xout=LeakyReLU(Conv1,1,1(Concatenate(xd∗,xh∗,xw∗)))

(7) y=LeakyReLU(Conv1,1,1(Concatenate(xres,xout)))+x.

Here, y represents the output feature map of the multi-dimensional multi-scale extraction module, and x is the input feature map. DWC denotes depthwise convolution, with subscripts indicating the corresponding kernel sizes. DG represents the depthwise gating mechanism. Res is the residual block. Concatenate denotes the concatenation operation. LeakyReLU is the activation function.

The input feature map x undergoes depthwise convolutions D(x), H(x) and W(x) to independently extract key features along the D, H, and W dimensions, while the residual block Res(x) extracts detailed information, resulting in four new feature maps xd,xw,xh,xres. Then, xd,xw,xh are transformed through a shared 1 × 1 × 1 convolutional mapping to obtain xd∗,xh∗,xw∗ enhancing inter-channel information interaction. The concatenate operation aggregates the multi-layer decoupled 3D spatial information to produce xout, which captures the variations in the image along the D, H, and W dimensions. This is concatenated with xres to enhance detailed information. Finally, the result is mapped through a 1 × 1 × 1 convolution and activated by LeakyReLU combined with the input x to prevent network degradation. This module, through its hierarchical decoupled multi-scale feature extraction and effective fusion strategy, is capable of capturing the complex positional variations of the pancreas and the subtle features of tumor regions.

Multi-dimensional decoupled channel attention module

The multi-dimensional decoupled multi-scale extraction module, while capable of extracting rich features, is prone to channel redundancy, which can affect segmentation results. The Squeeze-and-Excitation (SE) (Hu, Shen & Sun, 2018) module addresses this by modeling channel dependencies and automatically determining the importance of channels to enhance useful channels and suppress irrelevant ones. However, the SE module relies solely on global average pooling, resulting in a single feature representation that may lose local important features and lacks multi-scale understanding, making it difficult to capture multi-scale information such as the positional distribution of pancreatic organs in 3D data.

To address this issue, we propose a multi-dimensional decoupled channel attention module (structure shown in Fig. 3). This module can independently learn features from multiple dimensions, utilizing both global average pooling and global maximum pooling to capture average and salient feature responses. Specifically, the results from MDDMS are fed in parallel into three decoupled convolutions, generating multi-scale results across three dimensions. Subsequently, global average pooling and global maximum pooling are applied to each of the three scales, followed by shared PCEC (Pointwise Convolution with Expansion and Compression, ratio r = 4) processing. The two vectors from each dimension are summed to produce three vectors. Finally, through concatenation and a fully connected layer (FC), a weight vector containing channel importance information is generated. The mathematical expression for this module is as follows:

(8) xca_d,xca_h,xca_w=Conv5,1,1(xin),Conv1,5,1(xin),Conv1,1,5(xin)

(9) xca_d_out=PCEC(GMP(xca_d))+PCEC(GAP(xca_d))

(10) xca_h_out=PCEC(GMP(xca_h))+PCEC(GAP(xca_h))

(11) xca_w_out=PCEC(GMP(xca_w))+PCEC(GAP(xca_w))

(12) yca=xin×Sigmod(FC(Concatenate(xca_d_out,xca_h_out,xca_w_out)))+xin.

Here, xin is the input. Conv denotes the convolution module, with subscripts indicating the corresponding kernel sizes. PCEC stands for pointwise compression and expansion convolution. GAP represents global average pooling, GMP represents global maximum pooling, and FC denotes a fully connected layer.

Figure 3 The structure of Multi-dimensional Decoupled Channel Attention Module (MDCA).

(A) The MDCA module mainly consists of convolution, global average pooling, global max pooling, (B) PCEC (point convolution compression and expansion), and a fully connected layer.

The input xin undergoes three parallel convolutional computations, resulting in three outputs xca_d,xca_h,xca_w. Each output is subjected to global average pooling and global maximum pooling, yielding six tensors of shape (C, 1, 1, 1), representing the global average pooling and global maximum pooling tensors for each dimension. These tensors are then processed through a shared PCEC for channel compression and recovery mapping. Subsequently, the resulting vectors are pairwise summed to produce three learning vectors xca_d_out,xca_h_out,xca_w_out, which are then concatenated. The concatenated result is passed through a fully connected layer to generate channel importance information. Finally, the output is activated by a Sigmoid function to produce a probability vector, where the magnitude of the values represents the importance of the channels, thereby reducing the influence of irrelevant channels.

Multi-dimensional decoupled spatial attention module

The PC segmentation algorithm aims to achieve precise segmentation of the pancreas and its tumor regions in abdominal CT images. During the decoding phase, the gradual restoration of feature maps is crucial for the accuracy of the results. For 3D medical images, learning spatial structures is particularly important. Existing spatial attention models (SAM) (Woo et al., 2018) generate spatial attention maps by computing the average and maximum values along the channel dimension to highlight important spatial regions. However, SAM tends to overlook the relationships between dimensions in 3D medical images when calculating spatial attention, making it difficult to independently capture spatial information along the axial, coronal, and sagittal planes.

To address this issue, this study designs a MDSPA, structure shown in Fig. 4, aiming to independently capture spatial information from each dimension and aggregate it into a complete 3D spatial attention map to enhance the regions of interest. The RSA module first fuses high-level and low-level semantic features to concurrently capture both global context and fine-grained details - particularly critical for addressing tumor size variations. It then performs three parallel convolutions to independently extract spatial relationships across the three dimensions, explicitly modeling the inter-slice continuity that conventional 2D approaches struggle to capture. Following feature concatenation and aggregation, the module learns a spatial attention map that dynamically enhances key regions (such as blurred boundaries and micro-lesions) while suppressing irrelevant tissue interference, thereby providing implicit spatial guidance during the decoding phase. The process concludes with a convolution operation and residual connection to alleviate gradient vanishing in deep networks. The mathematical expression for the MDSPA is as follows:

(13) xspa_d,xspa_h,xspa_w=Conv5,1,1(xin),Conv1,5,1(xin),Conv1,1,5(xin)

(14) yspa=xin×Sigmod(Conv3,3,3(xspa_d+xspa_h+xspa_w)).

Here, xin represents the input feature map, Conv denotes convolution, with subscripts indicating the kernel size. Sigmod represents the activation function. yspa is the output of multi-dimensional spatial aggregation.

Figure 4 The overview of RSA block.

Our proposed (A) RSA (Residual Space Attention) block primarily consists of (B) MDSPA (multi-dimensional decoupled spatial attention), convolutions, InstanceNorm, and LeakyReLU.

The input xin undergoes three parallel convolutional computations to extract spatial information along the three dimensions, and the channels are compressed (compression ratio r = 4) to reduce interference from irrelevant channels, resulting in three outputs: xspa_d, xspa_h, and xspa_w. Subsequently, the three feature maps are aggregated through addition, and a 3 × 3 × 3 convolution is applied to map the channels back to their original size, ensuring spatial diversity and smoothness. The result is then mapped to a probability map using the Sigmod function and applied to xin to enhance the regions of interest, implicitly guiding the model to generate segmentation results. Through this design, the multi-dimensional decoupled spatial attention module can effectively capture spatial information across dimensions in 3D medical images, enhancing the model’s focus on regions of interest and thereby improving the accuracy of PC segmentation.

Experiment

Datasets

To validate the effectiveness of the proposed method, we employed two publicly available datasets: Medical Segmentation Decathlon Pancreas Tumour (MSDPT) and National Institutes of Health Pancreas (NIHP).

MSDPT dataset

The MSDPT dataset (Simpson et al., 2019) is a benchmark dataset for the pancreatic cancer segmentation task in the Medical Segmentation Decathlon challenge and is widely used in pancreas and tumor segmentation research. This dataset, provided by the Amber Simpson Cancer Center in the United States, includes a training set of 281 CT scans and a test set of 139 CT scans. The number of CT slices per scan ranges from 37 to 751, with an axial plane resolution of 512 × 512 pixels and a slice spacing ranging from 0.7 to 7.5 mm. The dataset contains three labels: background, pancreas, and tumor. The tumor category encompasses various pathological types, including intraductal mucinous neoplasms, pancreatic neuroendocrine tumors, and pancreatic ductal adenocarcinomas. Its inclusion of diverse tumor types and multi-center data makes it particularly suitable for evaluating model robustness. The dataset is publicly available at: http://medicaldecathlon.com/.

NIHP dataset

The NIHP dataset, provided by the National Institutes of Health Clinical Center, consists of 82 abdominal contrast-enhanced 3D CT scans. All scans were performed during the portal venous phase, approximately 70 s after intravenous contrast injection. The subjects included 53 males and 27 females, of whom 17 were healthy kidney donors, and the remaining 65 were individuals without significant abdominal diseases or pancreatic cancer lesions. The CT scan resolution is 512 × 512 pixels, with pixel size and slice thickness varying depending on the scanning equipment. The slice thickness ranges from 1.5 to 2.5 mm. Data acquisition was performed using Philips and Siemens MDCT scanners (120 kVp tube voltage). The pancreas was manually segmented slice-by-slice by a medical student and subsequently verified and corrected by an experienced radiologist to ensure annotation accuracy. The standardized acquisition protocol and rigorous annotation process establish it as a reliable reference for normal pancreatic anatomy. This dataset is accessible through The Cancer Imaging Archive (TCIA) at: https://www.cancerimagingarchive.net/collection/pancreas-ct/ or https://doi.org/10.7937/K9/TCIA.2016.tNB1kqBU.

Experimental environment and parameters

The experimental code is implemented in Python 3.10 and runs on an Ubuntu 22.04 environment. The hardware configuration includes an AMD Ryzen 7 5800X 8-Core Processor (3.80 GHz) CPU and an NVIDIA GTX 3060 (12G) GPU. The deep learning framework used is PyTorch 1.12.0, and the models are constructed with the assistance of the Monai 0.8.0 library. All experiments are conducted on the same device to ensure fairness. The implementation code for MDMU-Net is available at: https://github.com/SerendipityInTheWorld/MDMU_Net or Zenodo: https://doi.org/10.5281/zenodo.15714416.

The experimental parameters are set as follows: the batch size is set to 1, and the maximum number of iterations is 40,000. The loss function used is Dice Cross-Entropy Loss (DiceCEloss). To prevent overfitting, an early stopping mechanism is employed, where training is terminated early if the Dice coefficient on the validation set does not improve for 15 consecutive epochs, and the best-performing model on the validation set is saved for final testing. The AdamW optimizer is used, with an initial learning rate of 0.0001. All experiments employ a 5-fold cross-validation method to evaluate model performance.

Data preprocessing and augmentation

This study primarily uses the MSDPT dataset for training and testing, with the NIHP dataset serving as supplementary validation to assess the model’s generalization ability. Since the test set labels are not publicly available, all data splits are performed randomly based on the training set. The experiments employ a 5-fold cross-validation method, where the training set is randomly divided into five parts, with four parts used for training and one part for testing, repeated five times. The validation set is constructed by randomly sampling 1/8 of the allocated training data in each cross-validation.

During the data preprocessing stage, the training data is first resampled to unify the pixel spacing, optimizing the network training effectiveness. The resampled pixel spacing for MSDPT and NIHP datasets is [1.2, 1.2, 2.0] and [0.93, 0.93, 1.0], respectively. Next, the intensity values of the medical images are remapped to the range [0, 1] (the original intensity range for MSDPT and NIHP is [−145, 275]). Subsequently, foreground cropping is performed, and positive and negative samples are randomly cropped based on the labels, with a crop size of 96 × 96 × 96, to balance the ratio of positive and negative samples.

To enhance the model’s generalization ability and robustness, data augmentation operations are applied to the training data, including: (1) Random intensity adjustment: With a probability of 0.5, a random offset within the range [−0.1, 0.1] is added to the intensity value of each pixel.

(2) Affine Transformation: Each image undergoes mandatory rotation (about the Z-axis, ranging from 0 to π/30 radians) and uniform scaling (with scaling factors varying up to ±10% in each spatial dimension).

(3) For the validation and test sets, only resampling, intensity remapping, and foreground cropping are performed to maintain data consistency and authenticity. All preprocessing and data augmentation operations are implemented using the Monai framework.

Evaluation metrics

This study focuses on the segmentation of the pancreas and its tumors and employs the following evaluation metrics: Dice Similarity Coefficient (DSC), Jaccard Index (JC), sensitivity, specificity, and 95% Hausdorff distance (HD95). For the NIH dataset, the primary evaluation metrics are DSC, JC, and HD95. (1) DSC

The DSC is one of the most commonly used evaluation metrics in medical image segmentation. It measures the overlap between the ground truth (GT) and the prediction (PR). The formula for DSC is: (15) DSC(A,B)=2|A∩B||A|+|B|.

(2) JC

The JC represents the ratio of the intersection to the union of two sets. The formula for JC is: (16) JC(A,B)=|A∩B||A∪B|.

For both DSC and JC, A represents the ground truth, and B represents the prediction.

(3) Sensitivity

Sensitivity (also known as recall) measures the proportion of actual positive cases that are correctly identified as positive. The formula for sensitivity is: (17) Sensitivity=TPTP+FN.

(4) Specificity

Specificity measures the proportion of actual negative cases that are correctly identified as negative. The formula for specificity is: (18) Specificity=TNTN+FP.

For sensitivity and specificity: TP (true positives): The number of samples that are actually positive and correctly identified as positive. FN (false negatives): The number of samples that are actually positive but incorrectly identified as negative. TN (true negatives): The number of samples that are actually negative and correctly identified as negative. FP (false positives): The number of samples that are actually negative but incorrectly identified as positive.

(5) 95% Hausdorff Distance (HD95)

The Hausdorff Distance measures the maximum distance between two point sets. HD95 is a variant that considers the 95th percentile of these distances, thereby reducing the impact of outliers. The formula for HD95 is: (19) HD95(A,B)=95thpercentileof(max(supa∈Ainfb∈B∥a−b∥,supb∈Binfa∈A∥a−b∥))

where ∥a−b∥ represents the Euclidean distance between points a and b, sup denotes the supremum (the maximum of all possible distances), and inf denotes the infimum (the minimum of all possible distances).

Results and analysis

This section provides a detailed analysis of the experiments conducted to verify the efficiency and effectiveness of the proposed model. Ablation studies describe the impact of individual modules on the network’s segmentation results. This study primarily focuses on pancreatic cancer segmentation, with the pancreas serving as a target to further validate the model’s effectiveness and generalization ability. Comparisons with existing 3D segmentation models are also performed to confirm the proposed model’s efficacy.

Ablation analysis

To evaluate the effectiveness of our proposed MDMU-Net, we conducted comprehensive ablation experiments. Using 3D U-Net as the baseline, we designed five ablation experiments: (1) 3D U-Net, (2) 3D U-Net + Res, which retains the original structure but replaces skip connections and decoder parts with residual blocks, (3) 3D U-Net + MDDMS + Res, (4) 3D U-Net + MDDMS + MDCA + Res, and (5) MDM-UNet.

The experimental results (as shown in Table 1) indicate that the model’s performance in both pancreas and tumor segmentation tasks significantly improves with the gradual addition of modules.

Table 1 Ablation analysis of the proposed method on MSDPT and NIHP datasets.

Model	MSDPT (DSC)↑	MSDPT (HD95)↓	NIHP (DSC)↑	NIHP (HD95)↓	
Pancreas	Tumor	Pancreas	Tumor	Pancreas	Pancreas	
(1)	0.6606 ± 0.0199	0.2112 ± 0.0914	14.0012 ± 8.7226	63.1915 ± 28.0851	0.6738 ± 0.0570	37.7600 ± 17.1593	
(2)	0.6607 ± 0.0545	0.2635 ± 0.0892	55.5884 ± 34.1111	88.6984 ± 18.9174	0.7327 ± 0.0375	49.6844 ± 19.9717	
(3)	0.6804 ± 0.0432	0.2979 ± 0.1089	76.6021 ± 63.5840	91.5002 ± 53.5655	0.7478 ± 0.0364	52.1750 ± 29.9043	
(4)	0.6899 ± 0.0114	0.3017 ± 0.1031	58.7936 ± 24.0691	76.3987 ± 16.1543	0.7532 ± 0.0156	64.5968 ± 22.9676	
(5)	0.7108 ± 0.0320	0.3312 ± 0.1111	15.3417 ± 10.4495	49.5939 ± 22.4076	0.7709 ± 0.0264	15.3269 ± 7.3830	
Note:

Bold values indicate the optimal value of each evaluation index.

On the MSDPT dataset, the baseline model (3D U-Net only) achieves a pancreas DSC of 0.6606, a tumor DSC of 0.2112, a pancreas HD95 of 14.0012, and a tumor HD95 of 63.1915. After adding residual connections (res), the pancreas DSC shows little change, while the tumor DSC increases to 0.2635. However, the tumor HD95 significantly rises to 88.6984 ± 18.9174, indicating a decline in boundary segmentation accuracy. Further introducing the multi-scale feature fusion (MDDMS) module increases the pancreas DSC to 0.6804 and the tumor DSC to 0.2979, but the tumor HD95 further rises to 91.5002, suggesting poor stability in tumor boundary segmentation. After incorporating the multi-dimensional decoupled channel attention (MDCA) module, the pancreas DSC further improves to 0.6899, and the tumor DSC increases to 0.3017, while the tumor HD95 decreases to 76.3987, indicating that the channel attention module has a certain optimization effect on tumor segmentation performance. Finally, with the introduction of the spatial attention (MDSPA) module, the pancreas DSC reaches its highest value of 0.7108, the tumor DSC improves to 0.3312, and the tumor HD95 significantly decreases to 49.5939, demonstrating the significant role of the MDSPA module in enhancing tumor segmentation accuracy and boundary segmentation performance.

On the NIHP dataset, the baseline model achieves a pancreas DSC of 0.6738 and a pancreas HD95 of 37.76. With the gradual addition of modules, the pancreas DSC progressively increases to 0.7709, and the pancreas HD95 decreases to 15.3269, further validating the effectiveness of the module combination.

Overall, the MDDMS and MDSPA modules contribute the most to the improvement in pancreas and tumor DSC, while the channel attention and spatial attention modules significantly optimize the tumor HD95. The synergistic effects between the modules notably enhance the model’s segmentation performance in complex scenarios.

Model performance analysis

In this section, we compare our MDMU-Net with seven state-of-the-art segmentation methods on the MSDPT and NIHP datasets. The CNN-based models include: 3D U-Net (Çiçek et al., 2016) with its classic encoder-decoder architecture, Attention U-Net (Oktay et al., 2018) featuring attention gates, SegResNet (Myronenko, 2019) combining residual connections with multi-scale fusion, and DynUNet (Futrega et al., 2022) with dynamic architecture adjustment. The Transformer-based (Vaswani et al., 2017) approaches comprise: UNETR (Hatamizadeh et al., 2022b) integrating Transformers into the U-Net framework, nnFormer (Zhou et al., 2023) blending CNNs and Transformers, Swin UNETR (Hatamizadeh et al., 2022a) employing a Swin Transformer encoder, and 3DUX-Net using large-kernel depthwise convolutions to mimic Transformers.

The experimental results on the MSDPT dataset (Table 2) demonstrate that MDMU-Net achieves remarkable efficiency with only 26.97M parameters and 84.837G FLOPs. Specifically, compared to AttentionUNet (23.63M) with similar parameter counts, it improves tumor Dice by 0.0351 while reducing computations by 160.768G; when compared to UNETR (82.550G) with comparable computation costs, it uses 65.81M fewer parameters. This efficiency advantage originates from our depthwise convolution multi-dimensional decoupling design, which effectively eliminates redundant computations. The model delivers outstanding segmentation performance, achieving DSC scores of 0.7108 (pancreas) and 0.3312 (tumor), significantly surpassing competing methods. As shown in Table 3, its superior boundary segmentation accuracy is further validated by an HD95 score of 28.3127. These improvements stem from the synergistic combination of multi-dimensional decoupled multi-scale feature extraction modules and attention mechanisms, which precisely capture multi-scale features and demonstrate particular efficacy in small target segmentation.

Table 2 Comparison of parameters, floating point operations (FLOPs), category dice scores, and HD95 for different methods on the MSDPT dataset.

Model	Parameter (M)	Flops (G)	Dice↑	
Pancreas	Tumor	
3DUnet	7.91M	29.745G	0.6606 ± 0.0199	0.2113 ± 0.0914	
SegResNet	1.18M	15.430G	0.6832 ± 0.0351	0.2633 ± 0.0842	
3DUX-Net	53.0M	631.675G	0.6673 ± 0.0428	0.2889 ± 0.9659	
AttentionUnet	23.63M	245.605G	0.6698 ± 0.0561	0.2961 ± 0.1263	
UNETR	92.78M	82.550G	0.6148 ± 0.0496	0.1905 ± 0.0664	
nnFormer	149.25M	213.485G	0.6664 ± 0.0559	0.2668 ± 0.1224	
SwinUNETR	62.17M	329.331G	0.6742 ± 0.0359	0.2801 ± 0.0911	
Ours	26.97M	84.837G	0.7108 ± 0.0320	0.3312 ± 0.1111	
Note:

Bold values indicate the optimal value of each evaluation index.

Table 3 Comprehensive comparison of metrics for different methods on the MSDPT dataset.

Model	Dice↑	Iou↑	Sens↑	Spec↑	HD95↓	
3DUNet	0.4359 ± 0.0527	0.3223 ± 0.0393	0.4371 ± 0.0766	0.9997 ± 0.0001	38.5963 ± 18.2940	
SegResNet	0.4732 ± 0.0572	0.3536 ± 0.0492	0.4736 ± 0.0812	0.9997 ± 0.0001	37.6231 ± 15.6751	
3DUX-Net	0.4781 ± 0.0683	0.3554 ± 0.0566	0.4922 ± 0.0916	0.9997 ± 0.0001	36.3027 ± 22.6361	
AttentionUnet	0.4839 ± 0.0853	0.3608 ± 0.0703	0.5124 ± 0.1399	0.9997 ± 0.0001	36.8667 ± 18.2556	
UNETR	0.4017 ± 0.0547	0.2883 ± 0.0449	0.4025 ± 0.0854	0.9996 ± 0.0001	43.8937 ± 17.7325	
nnFormer	0.4666 ± 0.1123	0.3513 ± 0.0942	0.4628 ± 0.1328	0.9997 ± 0.0001	45.0921 ± 22.3324	
SwinUNETR	0.4781 ± 0.0663	0.3556 ± 0.0546	0.5055 ± 0.0902	0.9996 ± 0.0002	45.6697 ± 23.5726	
Ours	0.5210 ± 0.0651	0.4174 ± 0.0184	0.5521 ± 0.0836	0.9997 ± 0.0001	28.3127 ± 5.6131	
Note:

Bold values indicate the optimal value of each evaluation index.

Further comprehensive metric analysis (Table 3) reveals that MDMU-Net achieves optimal performance in Dice (0.5210), IoU (0.4174), and sensitivity (0.5521), with standard deviations lower than most comparative models, indicating more stable segmentation performance. The HD95 metric is 7.99 mm lower than the best comparative model (3DUX-Net), further confirming significant improvements in boundary segmentation accuracy. Notably, although SegResNet has the lowest parameter count (1.18M), its overall performance (Dice 0.4732) shows a significant gap compared to MDMU-Net, demonstrating that simply reducing parameters compromises model capability.

To validate the method’s generalization ability, experiments on the NIHP segmentation dataset (Table 4) show that MDMU-Net surpasses the current best method (DynUNet) in both pancreas segmentation Dice (0.7709) and IoU (0.6357), with average metric improvements of 6.4% and 9.0% respectively. Particularly in boundary accuracy, the HD95 (15.3267) is 6.387 mm lower than AttentionUNet, with the smallest standard deviation, indicating strong adaptability to different data distributions. The background class metric (Dice 0.9996) remains comparable to other models, showing that the method significantly improves small target segmentation while maintaining stability in large target segmentation.

Table 4 Comparison of dice and IOU metrics for different methods on the NIHP dataset.

Model	Dice↑	Iou↑	HD95↓	
Background	Pancreas	Mean	Background	Pancreas	Mean	Pancreas	
3DUnet	0.9995 ± 0.0001	0.6739 ± 0.0570	0.8367 ± 0.0286	0.9990 ± 0.0001	0.5220 ± 0.0642	0.7605 ± 0.0322	37.7600 ± 17.1593	
DynUNet	0.9995 ± 0.0001	0.7250 ± 0.0279	0.8623 ± 0.0140	0.9991 ± 0.0001	0.5832 ± 0.0287	0.7911 ± 0.0144	36.6011 ± 27.1476	
AttentionUnet	0.9996 ± 0.0001	0.7109 ± 0.0621	0.8552 ± 0.0311	0.9991 ± 0.0002	0.5681 ± 0.0720	0.7836 ± 0.0361	21.7137 ± 11.5158	
SwinUNETR	0.9995 ± 0.0001	0.7012 ± 0.0272	0.8504 ± 0.0136	0.9990 ± 0.0001	0.5545 ± 0.0316	0.7768 ± 0.0159	46.2683 ± 19.8101	
Ours	0.9996 ± 0.0001	0.7709 ± 0.0264	0.8852 ± 0.0132	0.9992 ± 0.0001	0.6357 ± 0.0306	0.8175 ± 0.0153	15.3267 ± 7.3831	
Note:

Bold values indicate the optimal value of each evaluation index.

We also present visualization results of multiple segmentations on CT slices (Figs. 5 and 6). For Sample2 in Fig. 5, the proposed MDMU-Net can still identify and accurately segment the tumor region. As shown in Sample3 in Fig. 6, the model also accurately segmented the pancreatic organ. However, in the pancreatic tumor segmentation of Sample5 in Fig. 5, the model shows slight deviations in localizing tumor boundaries, with some small tumor areas being missed. Although the model demonstrates effective segmentation capability for tumors and pancreatic organs in most samples, in Sample5, constrained by the complexity of tumor morphology in medical imaging (such as micro-lesions and blurred boundaries), the model exhibits insufficient sensitivity to fine structures during shallow feature extraction, or loses some detailed information due to down-sampling in deep networks, resulting in boundary deviations and missed segmentation. Furthermore, through the visualization analysis of the feature maps from the last layer of MDMU-Net (Fig. 7), we observe that the model exhibits high attention to the pancreatic region (yellow areas) and effectively identifies small-sized tumor targets.

Figure 5 Visual comparison of outputs from different models on the MSDPT dataset.

The red regions indicate the pancreas, and the green regions indicate the lesions.

Figure 6 Visual comparison of outputs from different models on the NIHP dataset.

The red regions indicate the pancreas.

Figure 7 Visualization of the output features from the last layer of MDMU-Net on the MSDPT and NIHP datasets.

The red area represents the pancreas, and the green area represents the lesions. Among them (A) is the feature visualization of the MSDPT dataset, and (B) is the feature visualization of the NIHP dataset.

Additionally, we use the paired t-test to calculate the p-values for the DSC metric to evaluate statistical significance. As shown in Tables 5 and 6, the p-values for MDMU-Net on the DSC metric are all less than 0.05, indicating that the performance improvement is statistically significant. The experimental results demonstrate that the proposed improvements in MDMU-Net are feasible and provide a new solution for 3D medical image segmentation.

Table 5 Statistical analysis using t-Test on the MSDPT dataset.

Model	Pancreas	Tumor	
DSC	p-value	DSC	p-value	
3DUnet	0.6606 ± 0.0199	0.010	0.2113 ± 0.0914	0.006	
SegResNet	0.4732 ± 0.0572	0.000	0.2633 ± 0.0842	0.035	
3DUX-Net	0.6673 ± 0.0428	0.007	0.2889 ± 0.1052	0.029	
AttentionUnet	0.6717 ± 0.0593	0.000	0.2961 ± 0.1263	0.025	
UNETR	0.6128 ± 0.0530	0.000	0.1905 ± 0.0684	0.004	
nnFormer	0.6664 ± 0.1058	0.039	0.2668 ± 0.1224	0.031	
SwinUNETR	0.6762 ± 0.0446	0.040	0.2113 ± 0.0914	0.037	
Ours	0.7108 ± 0.0320		0.3318 ± 0.1167		
Note:

Values in bold indicate the experimental results in the MDMU-Net.

Table 6 Statistical analysis using t-Test on the NIHP dataset.

Model	Pancreas	
DSC	P-value	
3DUnet	0.6738 ± 0.0570	0.003	
DynUNet	0.7250 ± 0.0278	0.022	
AttentionUnet	0.7109 ± 0.0621	0.041	
SwinUNETR	0.7012 ± 0.0272	0.003	
Ours	0.7709 ± 0.0264		
Note:

Values in bold indicate the experimental results in the MDMU-Net.

Discussion

This study proposes an MDMU-Net algorithm based on an improved 3D U-Net for the automatic segmentation of the pancreas and its tumors. The algorithm significantly enhances segmentation performance by introducing a multi-dimensional decoupled multi-scale feature extraction module, a multi-dimensional decoupled channel attention module, and a multi-dimensional decoupled spatial attention module. The multi-dimensional decoupled multi-scale module leverages the structural characteristics of 3D medical images to enhance feature representation capabilities. The multi-dimensional decoupled channel attention module improves the focus on key channels through multi-dimensional feature recalibration. The multi-dimensional decoupled spatial attention module fully utilizes the spatial information of 3D medical images to guide precise mask generation.

Through ablation experiments, we have validated the effectiveness of each module, particularly the significant improvements in pancreas and tumor segmentation tasks. Compared to existing advanced 3D segmentation models, MDMU-Net demonstrates superior performance on the MSDPT dataset, with DSC values of 0.7108 and 0.3312, outperforming other models in small tumor segmentation tasks. Additionally, the generalization capability of MDMU-Net has been verified on the NIHP dataset (DSC of 0.7709). Furthermore, we provide visual representations of some pancreas and tumor segmentation results, as shown in Figs. 5 and 6.

While our MDMU-Net demonstrates promising performance in pancreatic tumor segmentation through its innovative multi-dimensional decoupled architecture, several limitations warrant discussion. First, the current feature aggregation strategy relies on elementary concatenation operations, which may insufficiently capture cross-dimensional interdependencies between decoupled features. This simplification could potentially hinder the model’s ability to fully exploit complementary information across different anatomical planes. Second, the framework’s computational complexity increases proportionally with the decoupling granularity, posing challenges for real-time clinical applications. Third, the model’s performance on extremely small tumors remains suboptimal due to partial information loss during the multi-scale fusion process. Future work will focus on: (1) developing attention-guided dynamic fusion mechanisms to replace current aggregation approaches, (2) implementing hardware-aware model compression techniques, and (3) incorporating shape priori from clinical knowledge bases to enhance small lesion detection.

Conclusions

The MDMU-Net algorithm proposed in this study has achieved significant progress in the segmentation of the pancreas and its tumors. By combining multi-dimensional decoupled multi-scale feature extraction and multi-dimensional attention mechanisms, MDMU-Net effectively addresses the complex morphology and positional variations of the pancreas and its tumors, particularly excelling in small tumor segmentation tasks. The experimental results demonstrate that MDMU-Net achieves comprehensive advantages in pancreatic cancer CT image segmentation tasks. As shown in Table 3, our model significantly outperforms state-of-the-art comparison models (including 3DUX-Net, AttentionUNet, and SwinUNETR) across key metrics including segmentation accuracy (Dice/IoU), sensitivity, and boundary localization accuracy (HD95), with particularly notable improvements in both precision and stability for tumor boundary segmentation. These findings validate the application value of the proposed method in the field of medical image segmentation.

This study primarily focuses on multi-dimensional decoupled multi-scale and attention mechanisms for pancreas and tumor segmentation, treating information from each dimension equally without considering that the H and W dimensions in 3D medical image segmentation may contain more information. We plan to explore adjusting the decoupling strategy in future research to further enhance the performance of MDMU-Net.

Additional Information and Declarations

Competing Interests

The authors declare that they have no competing interests.

Author Contributions

Lian Lu conceived and designed the experiments, performed the experiments, analyzed the data, performed the computation work, prepared figures and/or tables, and approved the final draft.

Miao Wu conceived and designed the experiments, authored or reviewed drafts of the article, and approved the final draft.

Gan Sen performed the experiments, authored or reviewed drafts of the article, and approved the final draft.

Fei Ren conceived and designed the experiments, analyzed the data, prepared figures and/or tables, and approved the final draft.

Tao Hu conceived and designed the experiments, analyzed the data, prepared figures and/or tables, and approved the final draft.

Data Availability

The following information was supplied regarding data availability:

The code is available at GitHub and Zenodo:

- https://github.com/SerendipityInTheWorld/MDMU_Net.

- Lu, L., Wu, M., Sen, G., Ren, F., & Hu, T. (2025). MDMU-Net:3D Multi-dimensional Decoupled Multi-scale U-Net for Pancreatic Cancer Segmentation [Data set]. Zenodo. https://doi.org/10.5281/zenodo.15714416.

- The MSD Pancreas Tumour dataset is available at: http://medicaldecathlon.com/dataaws/

- The NIH Pancreas dataset is available at: https://www.cancerimagingarchive.net/collection/pancreas-ct.

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
