# Peer review of "MDMU-Net: 3D multi-dimensional decoupled multi-scale U-Net for pancreatic cancer segmentation"

_PeerJ Computer Science, doi:10.7717/peerj-cs.3059_

## Round 0.1 · original submission · Major Revisions

Please address all the requests and suggestions of the reviewers in detail.

·

Basic reporting

1. Please consider adding a blank space before brackets throughout the manuscript. For example, in Line 23, Change “MDMU-Net(Multi-” to “MDMU-Net (Multi-” and in Line 42, Change “6%(Siegel” to “6% (Siegel”
2. Typo in Line 123, change “Relate” to “Related”
3. Typo in Lines 141 and 144, change “pancreas's” to “ pancreas’ ''
4. Inconsistent formatting in Line 474, change “Results AND Analysis” to “Results and Analysis”
5. In Lines 192-193, explain the variables L, C, H, D, W
6. In Line 335, add a space after the comma. Change “map, Conv” to “map, Conv”
7. In Line 426, the phrase 'With a probability of 1.0' is confusing because it implies something uncertain, even though the rotation always occurs. Simply state that the image is always rotated.
8. In Line 520, the content of the 'Model Comparison' section seems out of place. Descriptions of prior models should be moved to the 'Related Works' section. This section should only briefly mention the state-of-the-art models used for comparison.
9. In Line 560, the data corresponding to the phrase in “Additionally, it achieves an HD95 score of 28.3127” is presented in Table 3, but it is mentioned while discussing the results of Table 2, which is confusing
10. Maintain consistent line spacing and add new lines before new sections or paragraphs when necessary. Examples:
- There is no new line before Line 455 “(4) Specificity” but there is a new line before Line 465 “(5) 95% Hausdorff Distance (HD95)”.
- Add line space before Line 474, which has the section header "Results AND Analysis"
- Lines 597 to 599 have 3 extra blank lines between the paragraphs, but other paragraphs within the same section, like Line 590, have only 1 extra blank line

Experimental design

The README.md file is empty in the GitHub source code link. Please add more details explaining the files, methods, and instructions to run the code and reproduce the results.

Validity of the findings

Lines 585 to 589 - no tables represent the data corresponding to these lines

Additional comments

The thorough ablation study and experimental analysis presented by the authors provide strong evidence that the proposed architecture improves segmentation performance for pancreatic cancer compared to state-of-the-art methods.

·

Basic reporting

Clear, professional English with technical accuracy.
Some technical terms need more explanation, ex. multi-dimensional decoupling
Needs a more comprehensive introduction of prior art and a specific knowledge gap.

Experimental design

Technically sound, and the scope is appropriate.
Use of 5-fold cross-validation is robust.

Validity of the findings

Results support a conclusion.
Adding Statistical significance testing would be valuable.
The ablation study is very comprehensive.

Additional comments

This study presents a novel architectural improvement to 3D U-Net segmentation for pancreatic cancer. For this, a highly lethal disease accuracy is needed for tumor delineation. The modular enhancement using spatial and channel attention provides a statistically and practically meaningful improvement in segmentation quality, particularly for small lesions. Thank you for such a nice article.

Please refer to peer review comments:
Abstract:
1. The abstract introduces MDMU-Net but lacks clarity on the novelty - explicitly state what differentiates it from other 3D U-Net derivatives.
2. You don’t have specific performance metrics from the results. You need to back up the claim “significantly outperforming other advanced models.”
3. Phrases “depthwise convolution” and “multi-dimensional decoupled” need brief clarification and simplification for general readers unfamiliar with the terms.
4. Can you explain the significance of the “Medical Segmentation Decathlon Pancreas Tumour” and “NIHP” datasets?

Introduction:
1. Can you clearly state what gap in pancreatic cancer segmentation prior works, ex. U-Net, Swin UNETR, fail to address?
2. You mentioned “only about 10% of patients are diagnosed at an early stage” – do you have numerical evidence to back the statement?
3. Since this study is pivotal on "multi-dimensional decoupling, the Introduction should briefly explain this term for clarity.
4. Can you explain why current models underperform on small tumor segmentation? Were the tumors missed? Or it could be poor boundary detection?
5. How do you connect segmentation performance to patient outcomes or treatment planning? Can you expand on this?

Methodology:
1. Can you add proper justification for modular design? Ex. What is the role of RSA, MDCA, and MDSPA in addressing specific segmentation challenges?
2. You are describing five ablation experiments, can you clearly define what is changed between models, ex. What is added from 3 -> 4?
3. For equations 13 and 14, can you add a description of each symbol, ex. Xin, σ, and how it is used in a pipeline?
4. You are missing hyperparameter details, ex. Specify the optimizer, learning rate, number of epochs, and loss function used.
5. For 5-fold CV strategy – describe whether stratification was used and how the final model was selected.

Experimental Results and Analysis:
1. While the improvements in Dice and HD95 scores are mentioned, no statistical significance tests are conducted. T-tests are reported. Add p-values or confidence intervals.
2. Can you do error analysis - discuss where and why the model fails?
3. Is there any justification why models like SwinUNETR or 3DUX-Net were selected as baselines? Do they have architectural similarity or SOTA performance?
4. You mentioned “outperforms existing advanced models” – you may need to explain which metrics are better and which models were outperformed.
5. Reduction in HD95 for tumors is impressive, but what does it mean in the clinical context? What is the significance of this?

Conclusion:
1. You claimed “broad potential for clinical application” – this seems overclaiming. You need to be realistic and specific when it comes to the potential for clinical application and validation.
2. Can you discuss the limitations of this work in a better context? Right now, you briefly mention “simplistic aggregation.”?
3. For future work, you mention “feature fusion”, can you please be more specific, ex. Integration with Transformer-based encoders or real-time deployment strategies, etc.
4. What is the relevance of this study, ex. How does the improved tumor segmentation aid oncologists or early diagnosis?

---

## Round 0.2 · accepted · Accept

Congratulations, and thank you for addressing all the concerns and suggestions of the reviewers.

·

Basic reporting

no comment

Experimental design

no comment

Validity of the findings

no comment

·

Basic reporting

This version looks better, thanks for addressing review comments.

Experimental design

This version looks better, thanks for addressing review comments.

Validity of the findings

This version looks better, thanks for addressing review comments.

Additional comments

This version looks better, thanks for addressing review comments.